# Are Non-Invasive Modalities for the Assessment of Atherosclerosis Useful for Heart Failure Predictions?

**DOI:** 10.3390/ijms24031925

**Published:** 2023-01-18

**Authors:** Kazuhiro Osawa, Toru Miyoshi

**Affiliations:** 1Department of General Internal Medicine 3, Kawasaki Medical School General Medical Center, Okayama 700-0821, Japan; 2Department of Cardiovascular Medicine, Okayama University Graduate School of Medicine, Dentistry and Pharmaceutical Sciences, Okayama 700-8558, Japan

**Keywords:** heart failure, coronary artery calcification, ankle-brachial index, carotid intima-media thickness, cardio-ankle vascular index

## Abstract

Heart failure (HF) is becoming an increasingly common issue worldwide and is associated with significant morbidity and mortality, making its prevention an important clinical goal. The criteria evaluated using non-invasive modalities such as coronary artery calcification, the ankle-brachial index, and carotid intima-media thickness have been proven to be effective in determining the relative risk of atherosclerotic cardiovascular disease. Notably, risk assessments using these modalities have been proven to be superior to the traditional risk predictors of cardiovascular disease. However, the ability to assess HF risk has not yet been well-established. In this review, we describe the clinical significance of such non-invasive modalities of atherosclerosis assessments and examine their ability to assess HF risk. The predictive value could be influenced by the left ventricular ejection fraction. Specifically, when the ejection fraction is reduced, its predictive value increases because this condition is potentially a result of coronary artery disease. In contrast, using these measures to predict HF with a preserved ejection fraction may be difficult because it is a heterogeneous condition. To overcome this issue, further research, especially on HF with a preserved ejection fraction, is required.

## 1. Introduction

Heart failure (HF) is the leading cause of morbidity and mortality worldwide, and its incidence is dramatically increasing with an increase in the aging population [1]. HF prevention relies on the identification of individuals at a high risk of HF. Traditional risk factors can be used to predict the 10-year incidence of HF [2]. However, risk assessment models based on traditional risk factors do not have a sufficient predictive value for HF prevention [3] because they do not include time-dependent risk exposures such as genetic or environmental factors [4]. Therefore, more accurate risk assessment modalities, which are non-invasive, low-cost, reproducible, and safe, are needed to identify individuals at a high risk of HF. Non-invasive modalities are used for the risk assessment of atherosclerosis in cardiovascular diseases, including HF, in a clinical setting. Recently, numerous prospective studies have investigated the predictive value and efficiency of an HF risk determination in asymptomatic populations. These non-invasive modalities include the coronary artery calcium (CAC) score [5,6] as a marker of coronary atherosclerosis, the ankle-brachial index (ABI) as an indicator of peripheral artery disease [7], the cardio-ankle vascular index (CAVI) as a marker of arterial stiffness, and the carotid intima-media thickness (IMT) as a measure of carotid atherosclerosis [8,9] (Figure 1). This article aims to describe key data from the current literature that support the association between these non-invasive modalities and for the prediction of HF.

## 2. Coronary Artery Calcification

CAC is a known independent risk predictor of atherosclerotic cardiovascular disease (ASCVD), including HF [10]. The CAC score is the most effective non-invasive measurement of atherosclerosis that can readily predict and identify ASCVD [11]. Individuals with higher CAC scores are at a higher risk of HF than those with lower CAC scores. Inverse logic can also be applied because individuals with HF have higher CAC scores than those without HF [6]. Adding CAC scores to traditional risk factors could improve their predictive accuracy for future HF incidence.

Several studies have demonstrated that CAC scores are associated with an increased risk of new-onset HF, after adjusting for standard risk factors. For example, in the *Rotterdam Study*, 1897 older adults without a history of ischemic heart disease or HF underwent CAC scoring and were followed up for the occurrence of HF and ischemic heart disease. Compared with a low CAC score (0–10), the adjusted hazard ratio (HR) of a high CAC score (>400) for having experienced HF was 4.1 (95% confidence interval (CI): 1.7–10.1) [5]. Moreover, CAC progression could be predictive of the incidence of HF. In the *Multi-Ethnic Study of Atherosclerosis* (MESA) study, a National Institute of Health-sponsored prospective population cohort study reported 182 HF events out of a total of 5644 participants (3.2%) during a median follow-up period of 9.6 years [12]. A CAC progression of 10 units per year was associated with a 3% increase in HF risk, independent of overt ischemic heart disease [12]. Evidently, the presence and severity of CAC are highly associated with a future HF incidence. However, the discriminative value of the CAC score in screening for HF remains controversial. In the *Rotterdam Study*, adding the CAC score to the cardiovascular risk factors resulted in an optimism-corrected increase in the c-statistic from 0.705 (95% CI: 0.666–0.754) to 0.734 (95% CI: 0.698–0.770) and substantially improved the risk classification of the participants (continuous net reclassification index: 34.0%) [5]. In contrast, the *Heinz Nixdorf Recall Study*, a population-based cohort study of 4814 asymptomatic individuals aged 45–75 years, demonstrated that adding the CAC score to the Framingham risk factors slightly increased the area under the receiver operating characteristic curve from 0.80 to 0.81 [6] Therefore, the predictive value of CAC should be carefully assessed considering the etiology of HF. According to official guidelines, HF is classified into four types [1]. In clinical practice, however, HF is usually treated and divided according to the left ventricular ejection fraction status into the following two types: HF with a preserved ejection fraction (HFpEF), characterized by a left ventricular ejection fraction of >50%; and HF with a reduced ejection fraction (HFrEF), characterized by a left ventricular ejection fraction of <40% [1]. The predictive and discriminative values of the CAC score are promising in HFrEF owing to its ability to predict ischemic heart disease, a major cause of HFrEF [13,14,15]. However, the discriminative value of CAC in HFpEF requires further discussion because HFpEF represents a heterogeneous syndrome derived from variable contributions of several pathophysiological processes [16].

Numerous studies have demonstrated an association between the CAC score and left ventricular diastolic dysfunction [17,18,19,20]. We previously reported that participants with higher CAC scores were more likely to have left ventricular diastolic dysfunction, suggesting that CAC is a risk predictor for future HFpEF [17]. Similarly, in a multi-center study of prospective coronary artery risk development in young adults comprising 3189 Caucasian and African American participants who were healthy at the baseline, Yared et al. reported that a higher CAC score in middle age was associated with a higher left ventricular mass and volume and left ventricular diastolic dysfunction [18]. These results clearly demonstrate the association between the CAC score and left ventricular diastolic dysfunction. However, the association between the CAC score and HFpEF remains controversial. In the MESA study, the association between the CAC score and HFpEF risk was investigated; a CAC score of >300 was significantly associated with HFpEF in contrast with a CAC score of 0 (adjusted HR: 1.68; 95% CI: 1.00–1.83). However, this association was only significant in women (HR: 2.8; 95% CI: 1.32–6.00) but not in men (HR: 0.91; 95% CI: 0.46–1.82) [21]. In addition, a relationship between the CAC score and left ventricular hypertrophy has been reported in asymptomatic patients with hypertension. Altunkan et al. demonstrated that the mean CAC score in patients with concentric left ventricular hypertrophy was significantly higher than that in patients with normal remodeling (315.4 ± 760.6 and 50.9 ± 187.4, respectively; *p* < 0.001), and the presence of CAC was associated with left ventricular hypertrophy (odds ratio [OR]: 2.68; 95% CI: 1.60–4.48; *p* = 0.001) [22]. Unfortunately, there have been only a few investigations on the CAC score and HFpEF. Additional studies are required to provide definitive conclusions regarding the association between the CAC score and HFpEF. The power of CAC as a risk predictor for HF is summarized in Table 1.

Non-contrast CT for the assessment of the CAC score is also useful in identifying extracoronary sources of calcification such as valvular calcification and thoracic artery calcification. In particular, valvular calcification may enhance the ability of CAC to predict HFpEF. An investigation carried out by the MESA study revealed that the progression of valvular calcification at the mitral annular and aortic valves was associated with an increased risk of HFpEF, independent of the standard risk factors, including the CAC score [23]. By comparing participants with and without valvular calcification at sites 1 and 2, valvular calcification was associated with an adjusted HR of 1.62 (95% CI: 1.21–2.17) and 1.88 (95% CI: 1.14–3.09) for HF events over 15 years, respectively [23]. Interestingly, these associations were only observed in patients with HFpEF but not in those with HFrEF. Furthermore, in the MESA study, the progression of valvular calcification was associated with various indices of the left ventricular structure and function [23]. Even though the detection of valvular calcification is not able to improve ASCVD risk prediction beyond what is known for CAC [24], the detection of valvular calcification and a concurrent assessment of the CAC score may be a promising method to overcome the deficiencies present in the CAC score and thus improve the discrimination of HFpEF patients. Additional research is required to definitively determine the efficiency of this combined assessment.

## 3. Ankle-Brachial Index

The ABI is a simple, inexpensive, and non-invasive modality for peripheral artery disease and a subclinical atherosclerosis estimation [25]. The ABI has previously been associated with mortality and cardiovascular events; an ABI of <0.9 has been associated with a higher risk of mortality and ASCVD events [26]. The ABI is also a promising risk predictor of HF [27,28]. An abnormal ABI is a marker of arterial stiffness, which may contribute to the pathogenesis of HF even in the absence of clinically apparent ischemic heart disease. 

Several studies have demonstrated a positive association between a low ABI and HF [7,27,28,29]. In a biracial, population-based atherosclerosis risk in communities (ARIC) study, Wang et al. indicated that an abnormal ABI was significantly associated with an increased risk of HF, regardless of the baseline ASCVD status during a median follow-up period of 5.5 years [7]. A low ABI (≤ 0.9) was associated with a higher incidence of HF in individuals with and without a known history of ASCVD (adjusted HR: 7.12 and 2.23, respectively) when compared with those with a higher ABI of 1.11–1.20 [7]. Moreover, the ABI significantly improved the risk discrimination of HF beyond the traditional risk factors. The c-statistics for the prediction of HF significantly improved from 0.696 to 0.710 in patients without a history of ASCVD and from 0.606 to 0.672 in those with a history of ASCVD [7]. In the IMPACT–ABI (*Impressive Predictive Value Of Ankle Brachial Index For Clinical Long-Term Outcome In Patients With Cardiovascular Disease Examined By ABI*) study, Nishimura et al. reported that both a low ABI (≤ 0.9) and a borderline ABI (0.91–0.99) were associated with a future incidence of HF in patients without a known history of HF over a mean follow-up of 4.8 years [29]. In a multivariate Cox proportional hazard analysis, a low ABI and a borderline ABI were independent predictors of the incidence of HF, with an HR of 3.00 (95% CI: 1.70–5.28; *p* < 0.001) and an HR of 2.68 (95% CI: 1.35–5.34; *p* = 0.005), respectively [29].

However, whether the ABI can predict HFpEF remains unclear. Investigators from the MESA study reported an association between the ABI and the risk of HF stratified by the left ventricular ejection fraction by assessing a mean follow-up period of 14 years [28]. The HR of a low ABI (<0.90) for HFrEF was 2.02 (95% CI: 1.19–3.04; *p* = 0.01) in the multivariate model adjusted for the established risk factors. However, this association was not significant for HFpEF in either the univariate (HR: 1.90; 95% CI: 0.88–4.08; *p* = 0.10) or multivariate (HR: 0.67; 95% CI: 0.30–1.48; *p* = 0.32) models [28]. Meanwhile, in the IMPACT–ABI study, echocardiographic data gathered during the onset of HF revealed that the prevalence of HFpEF in the low ABI group was 76.0%, which was similar to one study suggesting a possible association between a low ABI and the development of HFpEF [29].

Similar to CAC, a low ABI indicates systemic atherosclerosis, revealing a clear association between a low ABI and HFrEF, whereas a high ABI is associated with arterial stiffening due to the calcification of the arterial wall [30], and possibly with HFpEF. Choi et al. evaluated the correlation between a high ABI and left ventricular hypertrophy in a general population cohort consisting of 8246 people aged ≥50 years [31]. An ABI of 1.10 to 1.19 and 1.20 to 1.29 was significantly associated with left ventricular hypertrophy (ABI: 1.10 to 1.19, OR: 1.35; 95% CI: 1.19–1.53 and ABI: 1.20 to 1.29, OR: 1.59; 95% CI: 1.31–1.92), and an ABI ≥ 1.30 was marginally associated with left ventricular hypertrophy (OR: 1.73; 95% CI: 0.93–3.22; *p* = 0.078), suggesting that a higher ABI may be associated with left ventricular hypertrophy. Additional studies that specifically focus on the association between a high ABI and the development of HFpEF are necessary. The power of ABI as a risk predictor for HF is summarized in Table 2.

## 4. Cardio-Ankle Vascular Index

The CAVI measures the overall arterial stiffness from the origin of the aorta to the ankle and is a promising modality for the assessment of HF risk [32]. Although pulse wave velocity has been used as a marker of arterial stiffness, its use is limited because of the influence of the blood pressure on its value. However, the value of the CAVI is independent of the blood pressure at the time of measurement, which is a major advantage over pulse wave velocity [33]. Arterial stiffness increases the central pulse pressure and left ventricular afterload. This was suggested by the findings of a study that reported an association between the pulse wave and left ventricular hypertrophy in patients with hypertension and a correlation between the plasma B–type natriuretic peptide levels and the pulse pressure [34].

Several previous studies have provided evidence of an association between arterial stiffness and the risk of HF using carotid–femoral pulse wave velocity [35]. A study of carotid–femoral pulse wave velocity demonstrated that in a community-based cohort of middle-aged to elderly individuals, a higher carotid–femoral pulse wave velocity was associated with an increased risk of HF [35]. The adjusted HR of standardized, transformed carotid–femoral pulse wave velocity was 1.29 per standard deviation (SD) increase (95% CI: 1.02–1.64; *p* < 0.01). Regarding the HF subtype, a higher standardized transformed carotid–femoral pulse wave velocity was associated with both HFpEF and HFrEF, although the findings were not statistically significant, in part owing to a modest number of HF events [35]. The CAVI values in a matched case-control study were significantly higher in patients hospitalized for HFpEF than those in the control group. High CAVI values (>10.0) were undeniably associated with the hospitalization of HFpEF patients, with an OR of 6.76 [36]. A prospective cohort study consisting of 2932 patients demonstrated that high CAVI values (>9.5) were associated with an increased incidence of HF and hospitalization compared with patients with low CAVI values (≤7.55) (crude HR 2.28; 95% CI: 1.42–8.01; *p* = 0.005) [37]. However, as the number of HF events was small (n = 21), more cohesive studies are required to confirm the association between a high CAVI and an increased risk of HF. The power of carotid–femoral pulse wave velocity and CAVI as a risk predictor for HF is summarized in Table 3.

The association between an abnormal CAVI and left ventricular dysfunction has been reported in several studies [38,39,40]. The close relationship between the CAVI, pattern, and ratio of early-to-late diastolic transmitral flow velocity (E/A) and deceleration time on Doppler echocardiography has been previously described [39,40]. The CAVI was negatively correlated with the E/A and positively correlated with the deceleration time in patients with ischemic heart disease with a preserved ejection fraction (≥ 55%) [39]. The ratio of the mitral peak velocity of early filling (E) to the early diastolic mitral annular velocity (e’) is an index of the left ventricular filling pressure, which significantly correlates with the CAVI [41,42]. Left ventricular hypertrophy, a marker of organ damage in hypertension, is an important intermediary between hypertension and HF. The left ventricular mass is strongly correlated with the pulse pressure, confirming the importance of pulsatile phenomena determined by arterial stiffness [43]. In this context, the CAVI was shown to be positively correlated with the left ventricular mass index [36]. As left ventricular hypertrophy and left ventricular diastolic dysfunction are the major structural changes observed in the hearts of patients with HF, the CAVI may be useful for predicting the onset of HFpEF. However, few studies have investigated the association between the CAVI and HFpEF development; therefore, a further investigation is required. 

## 5. Carotid Intima-Media Thickness

Numerous studies have consistently demonstrated the positive association between an increased carotid IMT and HF [8,44,45]. For example, in the ARIC study, Effoe et al. reported that an increased carotid IMT was a valuable and graded predictor of HF incidence over a median follow-up period of 20.6 years [8]. Compared with quartile 1, quartile 4 was predictive of HF (HR: 1.60; 95% CI: 1.37–1.87), even after fully adjusted models were used [8]. Another analysis from the ARIC study investigated the association between the carotid IMT and the incidence of HF based on the diabetic status, and revealed that the carotid IMT was a weaker predictor of HF incidence among individuals with type 2 diabetes mellitus than among those with impaired fasting glucose or normal fasting glucose (HR of 1.12 (95% CI: 1.05–1.21) per SD increase in carotid IMT for diabetes mellitus compared with an HR of 1.18 (95% CI: 1.11–1.25) for impaired fasting glucose and an HR of 1.27 (95% CI: 1.20–1.34) for normal fasting glucose) [44]. These results suggest that the carotid IMT does not add a considerable predictive value to HF beyond the risks already accounted for by diabetes in individuals with type 2 diabetes mellitus [44]. 

Moreover, an association between the carotid IMT, left ventricular hypertrophy, and left ventricular dysfunction has been reported in several studies [46,47,48]. In a cross-sectional study with 1515 participants aged 36–45 years, the logistic regression results indicated that the carotid IMT was significantly associated with a risk of left ventricular hypertrophy (OR: 67.670; 95% CI: 13.35–342.97; *p* < 0.05) [46]. Chahal et al. demonstrated that in 2279 participants from a London life sciences prospective population cohort study without clinical cardiovascular disease, the carotid IMT was independently related to a reduced e’, a marker of the left ventricular diastolic function [47]. Nakanishi et al. evaluated the association between the carotid IMT and left ventricular global longitudinal strain (LVGLS) and peak left atrial longitudinal systolic strain (PALS), a marker of subclinical left heart dysfunction [48]. In multivariable analyses, the carotid IMT was associated with abnormal LVGLS (adjusted OR: 1.33 per 1 SD increase in IMT; *p* = 0.003) as well as PALS (adjusted OR: 1.33 per 1 SD increase in IMT; *p* = 0.005), independent of the traditional cardiovascular risk factors, echocardiographic parameters (including the left ventricular ejection fraction, left ventricular mass index, and diastolic dysfunction), and pertinent laboratory parameters [48]. In a recent investigation, Aladin et al. reported an association between the carotid IMT and the risk of all HF subtypes with the assistance of MESA participants [9]. Each SD increase in the measured maximum IMT was associated with both HFrEF and HFpEF in the unadjusted (HR: 1.57; 95% CI: 1.43–1.73) and demographically adjusted (HR: 1.61; 95% CI: 1.47–1.77) models. However, the statistical significance disappeared in the fully adjusted models, which included other traditional HF risk factors (HR: 1.11; 95% CI: 0.96–1.28) and interim coronary artery disease events (HR: 1.13; 95% CI: 0.98–1.30) [9]. Based on these studies, the carotid IMT may be a relatively weak predictor of the risk of HF. The power of carotid IMT as a risk predictor for HF is summarized in Table 4.

Meanwhile, carotid artery ultrasonography can evaluate arterial distensibility [49], which may have an improved predictive value for HF. The combined measurement of the carotid IMT and carotid artery stiffness within one examination, evaluating both atherosclerosis and a physiological dysfunction, may further refine the HF risk calculation. Carotid arterial distensibility is a significant predictor of future cardiovascular disease and all-cause mortality [50]. A small study reported the predictive value of carotid artery distensibility for the first acute HF incidence [51]. However, few studies have examined the relationship between carotid artery distensibility and HF incidence in large, multi-ethnic populations. Similarly, a combination strategy utilizing non-invasive modalities measuring atherosclerosis and flow-mediated dilatation examining an endothelial dysfunction may constitute a promising approach that could improve the predictive value of such non-invasive modalities for HF. Further studies are required to confirm the clinical significance of these combination therapies. 

## 6. Conclusions

In this review, we have described how the non-invasive modalities for measuring atherosclerosis relate to the risk assessment of HF based on the currently available literature (Figure 2). In general, all of these modalities are promising risk predictors for a future HF incidence. However, their predictive value might be influenced by the left ventricular ejection fraction. Specifically, their predictive value appears to be high when assessing the risk of HFrEF (because this condition potentially results from ischemic heart disease) and low when used to predict HFpEF, possibly because HFpEF is a heterogeneous syndrome. To overcome this ambiguity, further research, particularly on HFpEF, is required. The combination of non-invasive modalities such as the CAC score, ABI, CAVI, and IMT, has the potential to improve the risk assessment models based on the traditional risk factors. Recently, artificial intelligence technology has been introduced as a powerful tool to learn the complex relationships between the risk predictors and the clinical outcomes from a representative sample [52]. Non-invasive modalities for the assessment of atherosclerosis integrated with artificial intelligence may improve the HF prediction. Further research is required to develop a more useful prediction model.

## Figures and Tables

**Figure 1 ijms-24-01925-f001:**
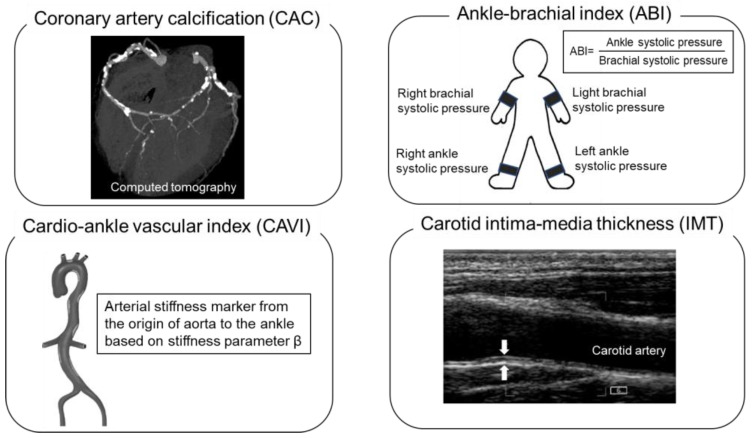
Non-invasive modality of atherosclerosis assessment.

**Figure 2 ijms-24-01925-f002:**
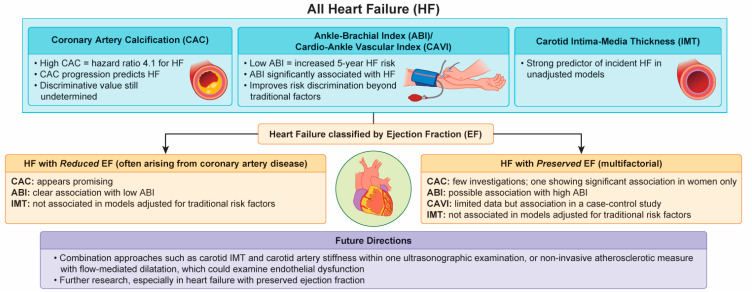
Clinical significance of non-invasive modalities used for assessment of atherosclerosis when evaluating the potential risk of heart failure.

**Table 1 ijms-24-01925-t001:** The power of coronary artery calcification as a risk predictor for heart failure.

First Author (Ref.#)	N	Mean Age (Years)	Male (%)	Follow-Up (Years)	Number of HF Events	Risk Categories: CAC	Reference: CAC	Adjusted HR
Leening et al. [5]	1897	70	42	6.8	78	>400	0–10	4.1 (1.7–10.1)
Kalsch et al. [6]	4814	65	44		105	Log(2)(CAC + 1)		1.07 (0.998–1.14) *
Bakhshi et al. [12]	5644	62	47	9.6	182	10 unit progression/year		3% increased risk
Sharma et al. [21]	6809	62	47	11.2	127 (HFpEF)	>300	0	1.68 (1.00–1.83)

HF: heart failure; HR: hazard ratio; HFpEF: heart failure with preserved ejection fraction; CAC: coronary artery calcification. * Odds ratio.

**Table 2 ijms-24-01925-t002:** The power of the ankle-brachial index as a risk predictor for heart failure.

First Author (Ref.#)	N	Mean Age (years)	Male (%)	Follow-Up (years)	Number of HF Events	Risk Categories: ABI	Reference: ABI	Adjusted HR
Wang et al. [7]	4160 without ASCVD	74	38	5.5		<0.90	1.11–1.20	2.23 (1.40–3.56)
						0.91–1.00		1.48 (0.91–2.42)
						1.01–1.10		1.03 (0.70–1.52)
						1.21–1.30		1.00 (0.61–1.63)
						>1.30		1.16 (0.67–2.00)
	843 with ASCVD	75	65	5.5		<0.90	1.21–1.30	7.12 (2.47–20.50)
						0.91–1.00		6.55 (2.24–19.17)
						1.01–1.10		4.81 (1.68–13.75)
						1.11–1.20		3.01 (1.06–8.58)
						>1.30		3.12 (1.00–9.73)
Gupta et al. [27]	13,150	45–64		17.7	1809	0.91–1.00	1.01–1.40	1.36 (1.17–1.59)
						<0.90		1.40 (1.12–1.74)
Prasada et al. [28]	6553	62	47	14	288 (all HF)	<0.90	1.01–1.40	1.22 (0.82–1.84)
						0.91–1.00		0.97 (0.66–1.43)
						>1.40		1.46 (0.59–3.60)
					155 (HFrEF)	<0.90	1.01–1.40	2.02 (1.19–3.04)
						0.91–1.00		0.95 (0.53–1.71)
						>1.40		2.59 (1.92–7.28)
					133 (HFpEF)	<0.90	1.01–1.40	0.67 (0.30–1.48)
						0.91–1.00		0.86 (1.48–1.55)
						>1.40		0.67 (0.09–4.91)
Nishimura et al. [29]	2824	69	72.3	4.8	105	<0.90	1.00–1.40	3.00 (1.70–5.28)
						0.91–0.99		2.68 (1.35–5.34)

HF: heart failure; HR: hazard ratio; ASCV: atherosclerotic cardiovascular disease; HFrEF: heart failure with reduced ejection fraction; HFpEF: heart failure with preserved ejection fraction.

**Table 3 ijms-24-01925-t003:** The power of carotid–femoral pulse wave velocity/cardio-ankle vascular index as a risk predictor for heart failure.

First Author (Ref.#)	N	Mean Age (years)	Male (%)	Follow-Up (years)	Number of HF Events	Risk Categories	Reference	Adjusted HR
Tsao et al. [35]	2539	64	44	10.1	170	Per SD unit increase in cfPWV		1.29 (1.02–1.64)
Miyoshi et al. [37]	2932	63	68	4.9	21	CAVI > 9.5	CAVI < 7.55	3.88 (1.42–8.01)

HF: heart failure; HR: hazard ratio; SD: standard deviation; cfPWV: carotid–femoral pulse wave velocity; CAVI: cardio-ankle vascular index.

**Table 4 ijms-24-01925-t004:** The power of carotid intima-media thickness as a risk predictor for heart failure.

First Author (Ref. #)	N	Mean Age (years)	Male (%)	Follow-Up (years)	Number of HF Events	Risk Categories: IMT	Reference: IMT	Adjusted HR
Effoe, et al. [8]	13,590	54	45	20.6	2008	0.62–0.69 mm	<0.62 mm	1.09 (0.93–1.28)
						0.70–0.79 mm	<0.62 mm	1.14 (0.97–1.33)
						>0.79 mm	<0.62 mm	1.60 (1.37–1.87)
						per SD increase in carotid IMT		1.20 (1.16–1.25)
Effoe, et al. [44]	7591 (normal fasting glucose)	54	39	20.6	843	per SD increase in carotid IMT		1.27 (1.20–1.34)
	4569 (impaired fasting glucose)	55	55.1	20.6	646	per SD increase in carotid IMT		1.18 (1.11–1.25)
	1430 (type 2 diabetes)	56	47.3	20.6	528	per SD increase in carotid IMT		1.12 (1.05–1.21)
Engstrom, et al. [45]	4691		13.2	13	75	per SD increase in carotid IMT		1.4 (1.2–1.8)

HF: heart failure; IMT: intima-media thickness; HR: hazard ratio; SD: standard deviation.

## Data Availability

Not applicable.

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
