# Peer review of "Are Non-Invasive Modalities for the Assessment of Atherosclerosis Useful for Heart Failure Predictions?"

_ijms, 2023, doi:10.3390/ijms24031925_

Round 1

Reviewer 1 Report

The paper provides a brief overview of the non-invasive modalities of atherosclerosis assessment, and examine their ability to assess the HF risk. The authors have discussed the challenges and provided the outlooks about future research focus and trends in this paper. The views reported here are meaningful and could provide important insights into the area of non-invasive modalities of atherosclerosis assessment, and HF risk asses. I would recommend its publication to IJMS after the authors addressing the following concerns.

1. The conclusion part is not rich enough. The authors need to clearly state the necessity of these researches and their final contribution.

2. The writing of the article needs some improvements. There are some basic language issues in the manuscript. The authors should check carefully and modify them.

Author Response

1. The conclusion part is not rich enough. The authors need to clearly state the necessity of these researches and their final contribution.

Response: We have revised the conclusion.

2. The writing of the article needs some improvements. There are some basic language issues in the manuscript. The authors should check carefully and modify them.

Response: The manuscript has been checked by professional native editing service.

Reviewer 2 Report

The manuscript may provide reason to e Han e attention toward clinical and instrumental signs  heralding pendono cardiovascular conditions

Author Response

Thank you for your comments.

Reviewer 3 Report

The authors have investigated and important cardiovascular theme regarding Heart failure prediction. Non-invasive modalities such as coronary artery calcification and carotid intima-media thickness have proven effective in determining relative risk of atherosclerotic cardiovascular disease. However, their ability to assess the HF risk has not been well established. Using these measures to predict HF is difficult due to heterogeneous condition. HF prediction systems are useful world wide and should be implemented and more accessible. Limitations are dictated by risk stratification in clinical settings and features. This review highlights the state of the art of general evidence of HF prediction by non-invasive assessment of atherosclerosis. 

The study is on point, well thought of and overall decently written. Introduction is sufficient for the purpose. The point by point exam is described in an according manner. 

My revision consists in the need in the conclusion for future prospective. This review should set for an incipit in future studies in adding new systems for HF prediction. The article would also greatly benefit from a summery table after each non-invasive modality chapter.

Author Response

We have revised the conclusion. In addition, we summary tables has been added after each chapter.

Round 2

Reviewer 3 Report

The authors have made the required adjustments and the paper is not fit for publication.